# Comparison of Soil Organic Carbon Stocks Evolution in Two Olive Orchards with Different Planting Systems in Southern Spain

José A. Gómez [1],*, Lizardo Reyna-Bowen [2], Pilar Fernández Rebollo [2] and María-Auxiliadora Soriano [3]

1    Institute for Sustainable Agriculture, Spanish National Research Council (IAS-CSIC), Alameda del Obispo S/N, 14004 Córdoba, Spain

2    Department of Forestry Engineering, University of Córdoba, Campus of Rabanales, Madrid-Cádiz Road Km. 396, 14014 Córdoba, Spain; lizardorb2021@gmail.com (L.R.-B.); ir1ferep@uco.es (P.F.R.)

3    Department of Agronomy, University of Córdoba, Campus of Rabanales, Madrid-Cádiz Road Km. 396, 14014 Córdoba, Spain; ag1sojim@uco.es

*    Correspondence: joseagomez@ias.csic.es

**Abstract:** This study presents an evaluation of soil organic carbon (SOC) and stock ($SOC_{stock}$) for the whole rooting depth (60 cm), spaced 55 months in two adjacent olive orchards with similar conditions but different tree densities: (i) intensive, planted in 1996 at 310 tree ha$^{-1}$; (ii) superintensive, planted in 2000 at 1850 tree ha$^{-1}$. This was carried out to test the hypothesis that olive orchards at different plant densities will have different rates of accumulation of SOC in the whole soil rooting depth. SOC increased significantly in the superintensive orchard during the 55-month period, from 1.1 to 1.6% in the lane area, and from 1.2 to 1.7% in the tree area (average 0–60 cm), with a significant increase in $SOC_{stock}$ from 4.7 to 6.1 kg m$^{-2}$. In the intensive orchard, there was not a significant increase in $SOC_{stock}$ in 0–60 cm, average of 4.06 and 4.16 kg m$^{-2}$ in 2013 and 2018, respectively. Results indicate a potential for a significant increase in SOC and $SOC_{stock}$ in olive orchards at higher tree densities when combined with temporary cover crops and mulch of chopped pruning residues. The increase is associated with an increase in SOC, mainly at a 0–15 cm depth. Results also point to the need for improve our monitoring capabilities to detect moderate increases in SOC.

**Keywords:** tree density; intensive orchard; superintensive orchard; deficit irrigation; bulk density

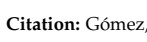



## 1. Introduction

Olive is a crop historically associated with the agricultural systems and cultural heritage of Mediterranean countries [1], being, in fact, one of the dominant crops in Mediterranean countries. Current statistics [2] show that approximately 97% of the world's acreage (10.3 Mha) is located in Mediterranean countries. Thus, olive is one of the main crops, in relative and absolute terms, in Spain, Tunisia, Morocco and Italy, covering 2.6, 1.6, 1.1 and 1.1 Mha, respectively. For this reason, any improvement on sustainability of olive cultivation might have a large national and regional impact. Traditionally agronomic practices for olive cultivation, historically a rainfed crop, have been aimed to optimize the use of rainfall water and the survival of the plantation in droughts. This means that a traditional olive plantation was based on low tree density, control of canopy size by regular pruning and soil management based on bare soil [3]. However, in recent decades, the increasing demand for olive oil, the access to water resources, the possibility of mechanized, or partially mechanized, harvesting, coupled with the discovery that deficit irrigation could boost olive yield [4] has resulted in a transformation of olive cultivation. This transformation has taken place during a period in which concerns regarding environmental damage associated with soil degradation, water erosion and offsite contamination led to a shift from bare soil based management (based on regular tillage and/or herbicide use) into more sustainable

management practices based on temporary cover crops during the rainy period and/or mulching with pruning residues [5]. As a result, there has been an increase in new olive plantations at higher plant densities, particularly if they have access to irrigation water, designed to be partially or fully mechanized for harvest. While traditional rainfed olive orchards may have around 80–120 tree ha$^{-1}$, trained with several trunks per tree, many new olive orchards are planted at a higher density and trained with a single trunk. Thus, the so-called intensive plantations are those with plant densities in the approximate range from 141 to 400 tree ha$^{-1}$. Another kind are superintensive plantations, which have an even higher plant density, usually above 1000 tree ha$^{-1}$, and trees are trained forming hedgerows, allowing for a fully mechanized harvest [6]. These more intensive plantations already represent a significant share of olive plantations in some producing countries, such as Spain, where 42.8% of the total olive acreage is planted at 141–400 tree ha$^{-1}$ and 6.1% are planted at more than 400 tree ha$^{-1}$ [7].

In the last years, there have been a number of studies aimed at determining the potential for increasing soil organic carbon (hereafter SOC) in olive orchards, many in relation to the use of cover crops. A relatively recent survey [8] in southern Spain demonstrated a great variability in SOC content in cover crop managed olive orchards, with the increase mainly concentrated in the topsoil. This study found SOC concentrations ranging from 0.5 to 2.5% in the top 15 cm, in accordance with the large variability observed in biomass production by cover crops along the lanes (between 0.65 to 2.53 t ha$^{-1}$ year$^{-1}$). All these studies measured SOC in the lanes, although it has long been described that SOC in olive orchards may change between the lanes and the area under the olive canopy area [3,9]. There are fewer studies on measuring soil OC stock (hereafter SOC$_{stock}$) in olive orchards also considering the subsoil. A study along a toposequence in a traditional regularly tilled rainfed olive orchard in Jaén (southern Spain) [10], conducted by using soil pits in the lanes, measured an average SOC$_{stock}$ of 6.6 kg m$^{-2}$ in the top 1 m soil depth without detecting any significant differences among the three different slope positions, summit, backslope and toeslope. An analogous study in a traditional olive orchard along a 430 m catena [11], also in the lanes of the orchard, found a significant increase in soil quality indicators (top 10-cm) in the downslope deposition area of the catena, but it did not find significant differences between different slope positions in SOC$_{stock}$ for the top 0.6 m of the soil, which was on average 3.90 kg m$^{-2}$. This contrast with a relatively recent study also along a toposequence in southern Spain, comparing the SOC$_{stock}$ in the top 1 m of the soil in the lanes, in two areas with two years of different soil management (spontaneous cover crops and chopped pruning residue mulching) [12]. In this study, the authors found a significant correlation with an increase in SOC$_{stock}$ along the toposequence, but no significant differences between management systems, with an average SOC$_{stock}$ of 4.03 kg m$^{-2}$. Another study carried out in central Italy [13] measured SOC$_{stock}$ in the top 0.9 m of the soil, as well in the lanes, in two olive orchards of different age (7 and 30 year) with permanent cover crop and the addition of chopped pruning residues, finding a significantly higher value for the older orchard, 13 vs. 7 kg m$^{-2}$. To the best of our knowledge there is only one published study in which the SOC$_{stock}$ of different typologies of olive orchards has been determined simultaneously [14]. In this study [14], SOC$_{stock}$ was measured in 22 olive orchards of different typologies (traditional, intensive and superintensive), albeit only in the top 0.3 m of the soil in the lane areas. This study reported a range of SOC$_{stock}$ from 0.38 to 4.02 kg m$^{-2}$ (average 2.66 kg m$^{-2}$). Based in the orchard age, since they measured only once in each orchard, they found statistically significant differences in the estimated SOC sequestration rate for different kinds of olive orchards: 0.02, 0.16 and 0.31 kg m$^{-2}$ year$^{-1}$, for traditional, intensive and superintensive orchards, respectively. This review suggests that the typology of the olive orchard may have a significant effect on soil SOC$_{stock}$, albeit confirmation is needed by additional studies, particularly aimed to measure at the whole tree rooting depth and, if possible, validating carbon sequestration rates with repeated measures on the same orchard. These future studies should also pay attention to the evaluation of the potential impact on SOC$_{stock}$ in olive orchards of the trees' effect on increasing soil organic carbon

content in their surrounding areas as compared to the lane areas. This tree impact has been demonstrated for analogous agroforestry systems in Mediterranean conditions [15,16]. However, to the best our knowledge, there are no published studies in this issue of tree influence in $SOC_{stock}$ in olive orchards, although there is a submitted manuscript based on measurements in a different orchard by our team, whose preliminary findings have been advanced in conferences [17], noting the relevance for this effect of the tree on SOC in the top soil. Improved empirical information on $SOC_{stock}$ in olive orchards in relation to their age and planting system can be extremely valuable to improve the contribution of this agricultural system to the "4 per 1000" initiative, to increase SOC in soils to compensate for global emissions of greenhouse gasses [18]. Furthermore, it can help to inform the appraisal for the sustainability of different olive cultivation systems, an issue very much in discussion in olive producing and consuming countries [19,20].

This manuscript presents a study aimed at testing the hypothesis that olive orchards at different plant densities will have different rates of accumulation of SOC in the whole soil rooting depth with two specific objectives: (i) evaluate the differences in SOC sequestration rate in the soil rooting depth between two olive orchards with different typologies, intensive and superintensive, in a period of five years, and (ii) appraise the effect of the differences in SOC concentration in the topsoil layer between the lane and the tree canopy area on the determination of $SOC_{stock}$.

## 2. Materials and Methods

### 2.1. Study Area and Olive Orchards Description

The study was carried out in a commercial olive tree farm, "El Alamillo", located in southwestern Spain (37°89'10.92″ N, 4°44'33.95″ W, Figure 1). It has a total extension of 73 ha, divided into several olive orchards with different planting system. Among them, we chose two nearby orchards spaced by approximately 80 m. One olive orchard with an extension of 6.2 ha, hereafter called intensive (I), was planted in 1996 at a tree density of 312 tree ha$^{-1}$ (4 × 8 m tree spacing, Picual variety), Figure 1. The other olive orchard, with an extension of 5.2 ha, hereafter called superintensive (SI), was planted in 2000 at a tree density of 1852 tree ha$^{-1}$ (1.35 × 4.0 m tree spacing, Arbequina variety) with the trees forming a hedgerow (Figure 1). In the superintensive orchard, the lanes were created in place without creating mounds where to plant the trees or any other soil movement. Both olive orchards were at the same elevation, approximately 146 m.a.s.l., with the intensive orchard having an average slope of 6.6% in the lane direction, while the superintensive orchard showed an average slope of 4.8% in the lane direction.

The study area had a Mediterranean-type climate, with an average (for 2000–2021) annual rainfall of 598 mm, of which 75% occurs from October to March. Average annual temperature is 17.6 °C, with a maximum average daily temperature in July of 27.8 °C, and minimum in January of 8.4 °C. The average annual cumulative potential evapotranspiration is 1408 mm, with daily average values ranging from 7.6 in July to 1.1 mm day$^{-1}$ in December. The soils in the farm are formed on Miocene marls and have been classified as Calcic Luvisols [21], according to the FAO classification.

Olive trees were deficit irrigated, using a drip irrigation system, from May to September with a total amount of irrigation water of approximately 120 mm per season. Harvesting is semi-mechanized in the intensive orchard using tree-shakers, and fully mechanized in the superintensive orchard using an over-the-row harvester similar to the ones used for trellis vineyards. In the intensive orchard, harvesting is carried out from mid-November to early January, while in the superintensive is usually in December, with the specific dates for a given year varying depending on weather conditions and fruit ripening. In both orchards, soil management is based on temporary cover crops managed by mowing combined with a mulch of chopped pruning residues (done annually) in the lanes, and herbicide used only in the tree lines.

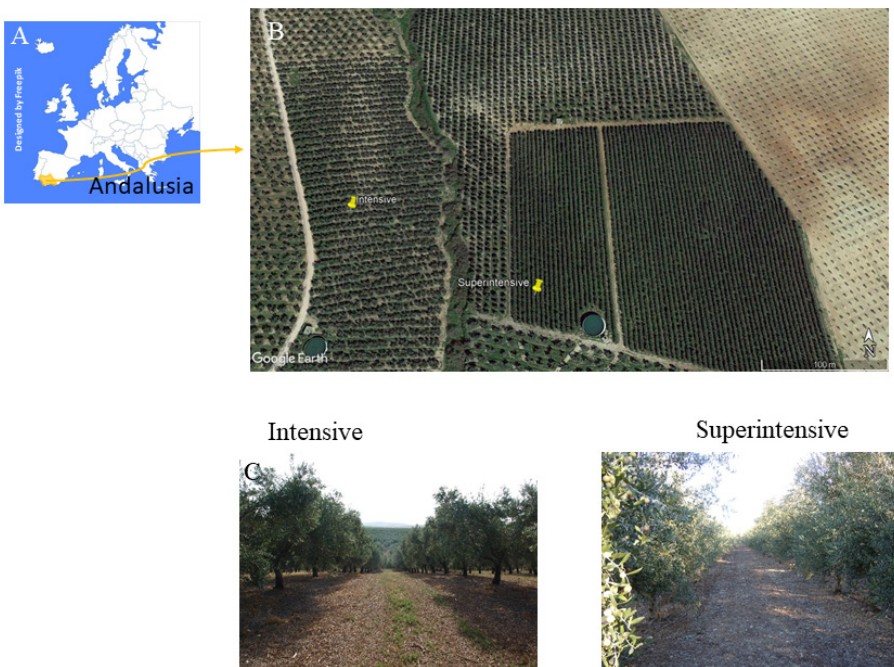

**Figure 1.** Location map (**A**), air view (**B**) and close view (**C**) of the two sampled olive orchards.

### 2.2. Soil Sampling

Soil sampling was carried out in both olive orchards twice, first in 27–28 November 2013 and secondly in 19–20 June 2018. In each orchard and sampling date, 16 soil sampling points were created in a transect perpendicular to the tree lines, encompassing three orchard lanes, which meant a 16 and 8 m transect for the intensive and superintensive orchard. Eight of these points were located in the lane area, and selected outside the tree canopy projection, while another eight were located in the area below the olive tree canopy projection. Samples in the tree and lane area were evenly spaced, which were obtained in areas within and outside the machine tyre traffic marks. Sampling was carried out in the two, lane and tree areas because prior studies have noted differences in bulk density and SOC in olive orchards [3,9]. Soil samples were obtained at four different depth intervals (0–5 cm, 0–15 cm, 15–30 cm and 30–60 cm, where possible) at each sampling point, having previously removed the grass and mulch from the surface. Soil samples were obtained combining a manual soil sampler (for the top 0–5 cm depth samples) and a hydraulic soil corer (Giddings®) with a 31.61 mm inner diameter. Sampling depths with the hydraulic corer were chosen according to the depths in which the corer allowed the obtaining of intact cores (usually >10 cm) to prevent contamination by soil corresponding to different sampling depths. Maximum soil sampling depth was decided based on the results for the maximum rooting depth observed in the analysis of soil cores obtained up to 1 m depth prior to the sampling carried out in November 2013. This estimation of maximum rooting depth was complemented with two soil profiles described by digging two pits, one in each orchard, with the pits starting near the tree trunk and crossing below and outside the tree crown projection area. The description of the soil profile, including visual assessment of root amount by root thickness, was carried out according the NRCS guidelines [22], which appear in Table 1. Locations of the sampling were recorded in 2013, in order to sample in the same area in the 2018 sampling.

**Table 1.** Soil properties from the soil profile descriptions in the soil pit created at each of the olive orchards in 2018. Letters A, B and C refer to soil horizon from the soil profile description. $C_{org}$: organic carbon (Walkley–Black method), $N_{org}$: organic nitrogen, $P_{exchang}$: exchangeable phosphorus, $K_{avail}$: available potassium, CEC: cation exchange capacity; coarse roots refer to those between 5–10 mm in diameter, very fine roots refer to those smaller than 1.0 mm in diameter. Note that depth for BC horizon refers to the maximum depth excavate (indicated by >).

| Olive Orchard | Soil Depth (cm) | Clay (%) | Silt (%) | Sand (%) | Coarse Roots | Very Fine Roots | $C_{org}$ (%) | $N_{org}$ (%) | $P_{exchang}$ (ppm) | $K_{avail}$ (ppm) | CEC (cmol$_c$ kg$^{-1}$) | pH (1:2.5 H$_2$O) |
|---|---|---|---|---|---|---|---|---|---|---|---|---|
| **Intensive** | | | | | | | | | | | | |
| A | 20–25 | 38.9 | 38.6 | 22.5 | Common | Common | 0.87 | 0.081 | 4.1 | 544 | 24.4 | 8.66 |
| B | 65–80 | 35.0 | 49.7 | 15.3 | Few | Few | 0.55 | 0.039 | 4.0 | 199 | 16.0 | 8.66 |
| BC | >85 | 26.8 | 61.0 | 12.2 | None | Very few | 0.32 | 0.017 | 3.1 | 87 | 14.1 | 8.81 |
| **Superintensive** | | | | | | | | | | | | |
| A | 20–30 | 36.4 | 26.3 | 37.3 | Common | Common | 0.83 | 0.090 | 4.5 | 384 | 25.3 | 8.24 |
| B | 65–80 | 36.5 | 37.2 | 26.3 | Few | Few | 1.03 | 0.089 | 5.0 | 233 | 26.9 | 8.03 |
| BC | >80 | 28.9 | 51.2 | 19.9 | None | None | 0.40 | 0.025 | 3.7 | 101 | 17.3 | 8.25 |

*2.3. Soil Analysis*

The soil samples were ground, passed through a 2 mm sieve, and homogenized. In the 2013 sampling, stoniness (i.e., coarse material >2-mm size) was determined as a fraction (*w/w*) of the total mass, in each soil sample. Additionally, in the 2013 sampling, a subsample was oven dried at 105 °C for 72 h until constant weight *to obtain soil sample moisture,* while the rest of the soil in each sample was air dried. Soil bulk density was calculated by dividing the dry weight by the volume of the soil sample (98.2 cm$^3$ in the case of the top 0–5 cm; 117.7 cm$^3$ in the sampling from 0–15 and 15–30 cm depth, and 235.4 cm$^3$ in the 30–60 soil depth sampling). Soil organic carbon (SOC) was determined by wet oxidation [22]. Soil organic carbon stock (SOC$_{stock}$) for each soil depth interval, and for the whole soil profile, was calculated according to [23], as:

$$SOC_{stock} = \sum_{i=1}^{n} SOC_i \, d_i \, BD_i \, (1 - stone) \times 1000 \qquad (1)$$

where SOC is soil organic carbon concentration (in fraction, 0–1), d is the sampling depth (m), BD is bulk density (t m$^{-3}$), and stone refers to the fraction (0–1) of coarse material larger than 2-mm in soil samples. Soil depths considered were 0–5, 0–15, 15–30 and 30–60 cm, and so we could calculate the cumulative SOC$_{stock}$ (kg OC m$^{-2}$) for 0–5, 0–15, 0–30 and 0–60 cm depth. For both sampling dates, 2013 and 2018, we used for Equation (1) the stoniness and bulk density determined in 2013 because we did not find, as expected, statistically significant differences in stoniness and bulk density in a subsample of soil cores obtained prior to the sampling of 2018. To convert the lane and tree SOC$_{stock}$ values, for the 0–60 cm soil depth, into a single orchard value, we calculated a weighted average, with the weighting value being the ground cover by the olive canopy at the time of sampling. This canopy cover was determined, based on the analysis of air images, as 0.32 and 0.68 for the tree and lane areas, respectively, in the intensive orchard, and as 0.5 for the lane and tree areas for the superintensive orchard. Since we did not find significant differences between 2013 and 2018, we used the same conversion values for both dates.

The other soil chemical determinations carried out for the soil profile description: exchangeable phosphorus, available potassium, cation exchange capacity, pH and organic N, were analyzed as described in MAPA [24], and the particle size analysis by the hydrometer method [25].

*2.4. Statistical Analysis*

Differences between average values of SOC and SOC$_{stock}$ in the two olive orchards or between the two sampling dates by depth and location within the orchard were evaluated

using a non-parametric Kruskal–Wallis test. All the statistical analyses were performed with STATA (ver. 15.1)

## 3. Results and Discussion

### 3.1. Comparison of SOC and SOC$_{stock}$ between Planting Systems in 2013

The comparison of SOC by soil depth and location (lane or tree) between the two olive orchards in the 2013 sampling date is summarized in Figure 2A. There was an overall trend towards higher average SOC (for all depths and locations) in the superintensive orchard, 1.10%, as compare to the intensive one 0.89% ($p < 0.05$, according to a Kruskal–Wallis test). Within each orchard, the intensive orchard presented a statistically significant higher average SOC in the average values for the tree area (0.97%, $p < 0.05$) as compared to the lane area (0.82%), while the superintensive orchard did not show these differences in SOC between the lane and tree areas, at 1.10%. Higher SOC in the tree area as compared to the lane area in olive orchards has been previously noted by several authors [3,17], as well as in other agroforestry systems [15,16]. In all these cases, this increase in SOC in the tree canopy projection area has been attributed to the increased biomass concentration (by roots and tree debris) closer to the tree. It worth noting that none of these studies [3,15–17] studied a planting density as dense as that of the superintensive, hedgerow-forming orchard in our study. In this case, no significant differences in SOC between the lane and tree areas were detected, which we attributed to the proximity of the olive trees, resulting in a more similar biomass contribution by the tree (from roots, tree debris as well as pruning residues) between the lane and tree area. Nevertheless, differences in SOC between the tree and lane areas sometimes take many years to develop in agroforestry systems [16], but it is apparent that in the 13 years since the planting of the superintensive orchard this has not been the case. Figure 2A also allows the identifying of soil depths where the differences in SOC between the intensive and superintensive orchards were concentrated. These tended to be concentrated in the top 0–15 cm soil depth, in both the lane and the tree area. These differences in SOC can be attributed to a higher biomass return to the soil by the superintensive orchard due to the higher number of trees, as suggested by the study of Lopez-Bellido et al. [14] when measuring SOC$_{stock}$ in the top 30-cm of the lane areas in olive orchards, also in southern Spain. The annual increase in SOC$_{stock}$ in the superintensive orchard is in the range observed by Lopez-Bellido et al. [14] and to that determined by Xiloyannis et al. [26] for a traditional olive orchard in southern Italy following a sustainable approach. It represents around 8% of the annual input in organic carbon to the soil in an olive orchard according to the calculations by Xiloyannis [27].

One necessary piece of information to determine SOC$_{stock}$ is soil bulk density (BD) is shown in Figure 3. As expected, BD was significantly higher in the lane area, with an average of 1.13 t m$^{-3}$, as compared to the tree area, 0.96 t m$^{-3}$ in both orchards, due to the machinery traffic [3,9]. However, we did not detect statistically significant differences in BD between the two orchards, nor did we detect statistically significant differences in soil coarse material >2-mm size, which was in average 10% in both olive orchards.

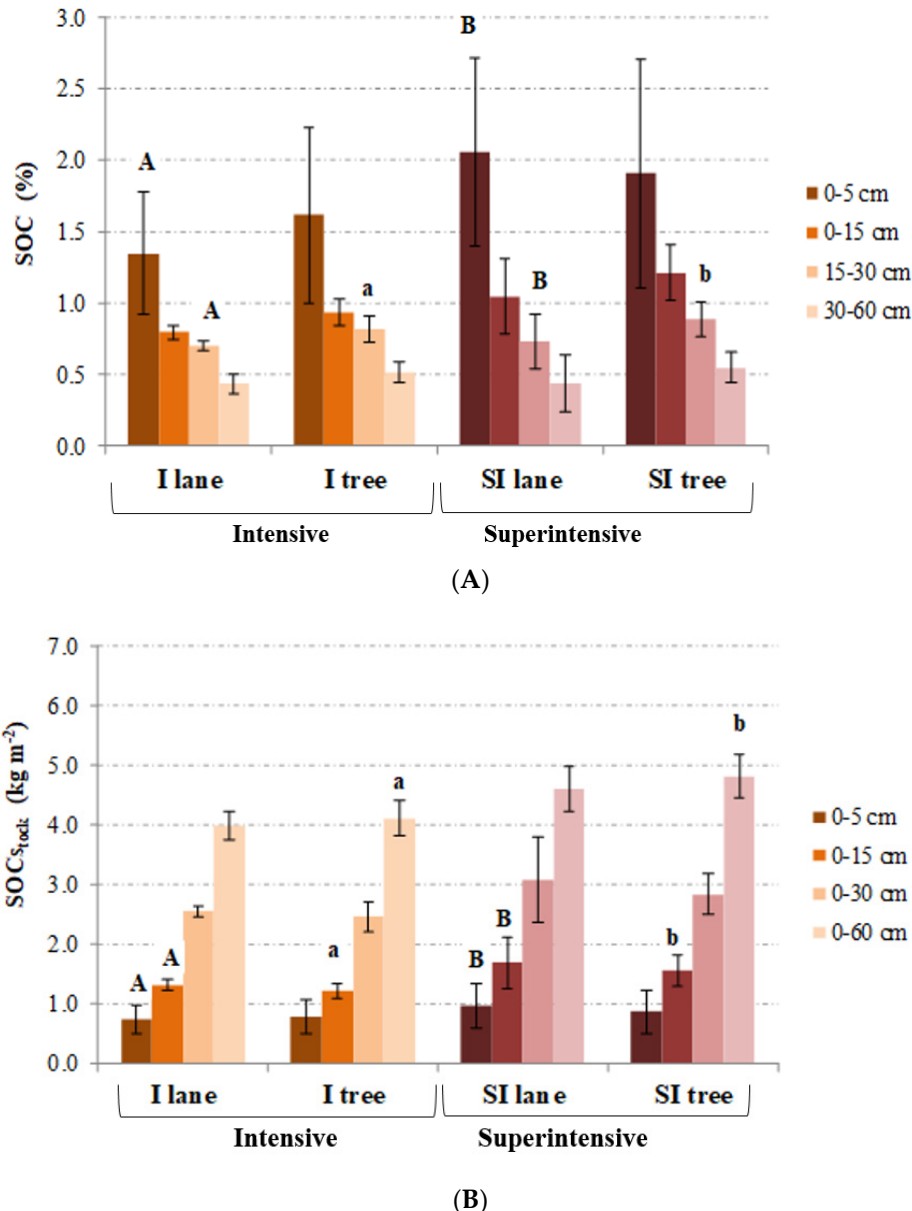

**(A)**

**(B)**

**Figure 2.** (**A**) Soil organic carbon concentration (SOC; %), and (**B**) cumulative soil organic carbon stock (SOC$_{stock}$; kg m$^{-2}$) by soil depth and location (lane or tree) within the two olive orchards (I: intensive; SI: superintensive) for the soil sampling carried out in 2013. Different letters indicate statistically significant differences ($p < 0.05$, using a Kruskal–Wallis test) between the two orchards for the same depth and location (lane, capital letters, or tree, lowercase letters). Error bars are standard deviation, n = 8.

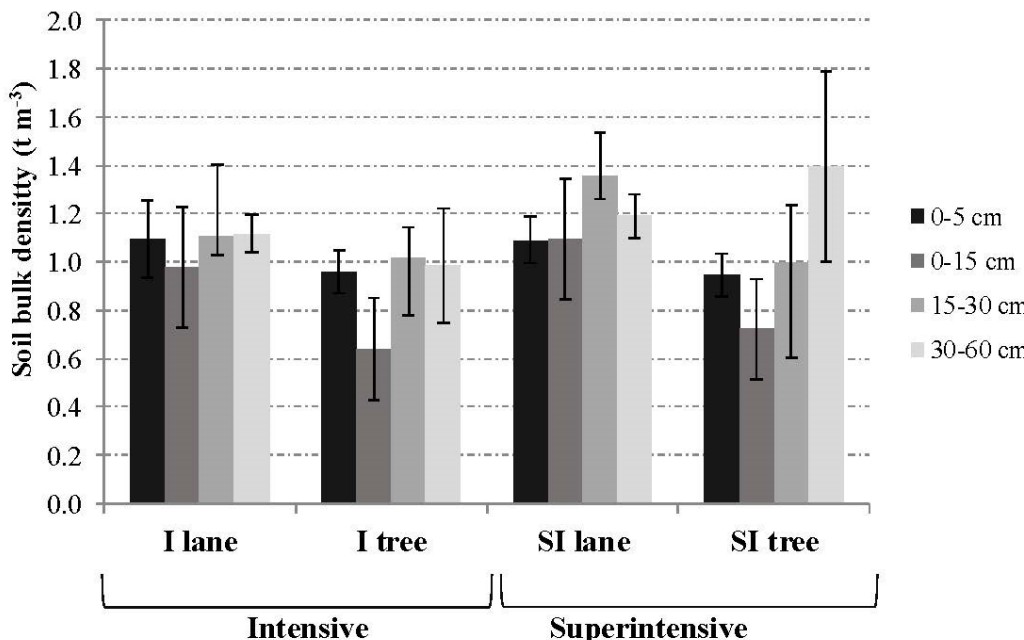

**Figure 3.** Soil bulk density by depth and location (lane and tree) within the orchard for the two sampled olive orchards (I: intensive; SI: superintensive). Error bars are standard deviation, n = 8.

Figure 2B summarizes the SOCstock by depth and location in both olive orchards. First, no statistically significant differences in SOCstock for the top 0–60 cm soil depth were observed between the lane and tree area within any of the two orchards, with the highest SOC values in the tree area as compared to the lane area, in the intensive orchard, compensated by a higher BD in the lane area. When comparing between planting systems by depth and location (Figure 2B), a significantly higher value in $SOC_{stock}$ is apparent in the tree area of the superintensive orchard than in that of the intensive orchard, in the soil depth of 0–15 and 0–60 cm. Nevertheless, when we integrated the two different areas of the orchards to obtain the $SOC_{stock}$ value for each olive orchard, no statistically significant differences were detected, with average (±SD) values for the whole 0–60 cm rooting depth of 4.03 (±0.29) and 4.47 (±1.01) kg m$^{-2}$ for the intensive and superintensive orchard, respectively. The lack of significance might be due to the relatively low number of replications available for the statistical analysis. The results for 2013 indicated that the effect of higher tree density, resulting in higher $SOC_{stock}$ in orchards, suggested by the study of Lopez-Bellido et al. [14], was not large enough to be detected, although the differences in mean $SOC_{stock}$ values between orchards suggests that it might be detected with a larger number of samples to deal with the spatial variability. The overall values obtained for $SOC_{stock}$ for the whole soil rooting depth (60 cm in our case see Table 1) were similar to those measured by Gómez et al. [20] in the top 0.6-m of a traditional olive orchard, but lower than those reported by Lozano-García et al. [10], who measured 6.6 kg m$^{-2}$ in the top 1 m soil depth, or by Massaccesi et al. [13], who measured approximately 7 and 13 kg m$^{-2}$ of soil organic carbon in the top 0.9-m of soil depending on the orchard age. The lower $SOC_{stock}$ values measured in our study can be explained to a large extent by the relatively shallow soil profile, 60 cm in average, with no visible tree roots below the 60 cm depth, in which the BC horizon started (Table 1). The chosen sampling depth, decided based on the soil available for the olive tree roots, complicates the comparison with studies carried out using other sampling depths, but it can provide insight, particularly when soil sampling is repeated over time, on the actual potential for soil organic carbon storage by vegetation in shallow soils.

### 3.2. Comparison of SOC and SOC$_{stock}$ between Planting Systems in 2018

The soil sampling carried out in 2018 allowed validation of the results based on the 2013 sampling, as well as identifying any possible trend in temporal changes in SOC. Figure 4A presents the values measured in 2018 of SOC by olive orchard, soil depth and location within the orchard. There is an overall trend, which will be fully discussed in the next section, towards an increase in SOC in the superintensive orchard as compared to its 2013 levels; while in the intensive orchard, the 2018 SOC values were similar to those measured in 2013. As a result, the superintensive orchard had statistically significant ($p < 0.005$) higher SOC as compared to the intensive orchard. Results for the 2018 sampling confirmed the same trend towards a statistically significant higher SOC in the tree area ($p < 0.05$) as compared to the lane area in the intensive orchard, and the lack of this differentiation in the superintensive orchard, whose reasons have been discussed in the previous section. It is apparent that this differentiation might never appear in a superintensive orchard, explained by the more homogeneous distribution of tree roots and tree debris and pruning residues across the orchard. Figure 4A also indicates the areas and depths where the differences in SOC between the intensive and superintensive orchard appears to spread, also encompassing the lane and tree areas below 30-cm soil depth.

Figure 4B summarizes the SOC$_{stock}$ by soil depth and location between both olive orchards, showing a similar trend to the one discussed for SOC above. In short, SOC$_{stock}$ in the intensive orchard in 2018 for the different depths tended to be quite similar to that measured in 2013, without statistically significant differences, and without differences between the lane and tree areas for the reasons already discussed in the previous section, corresponding to the 2013 results. The overall SOC$_{stock}$ for the intensive orchard was 4.16 ($\pm 1.12$) kg m$^{-2}$. In the superintensive orchard, there was an increase compared to 2013 as a result of the increase in SOC discussed in the previous paragraph, with an overall SOC$_{stock}$ for the superintensive orchard of 6.1 ($\pm 1.22$) kg m$^{-2}$.

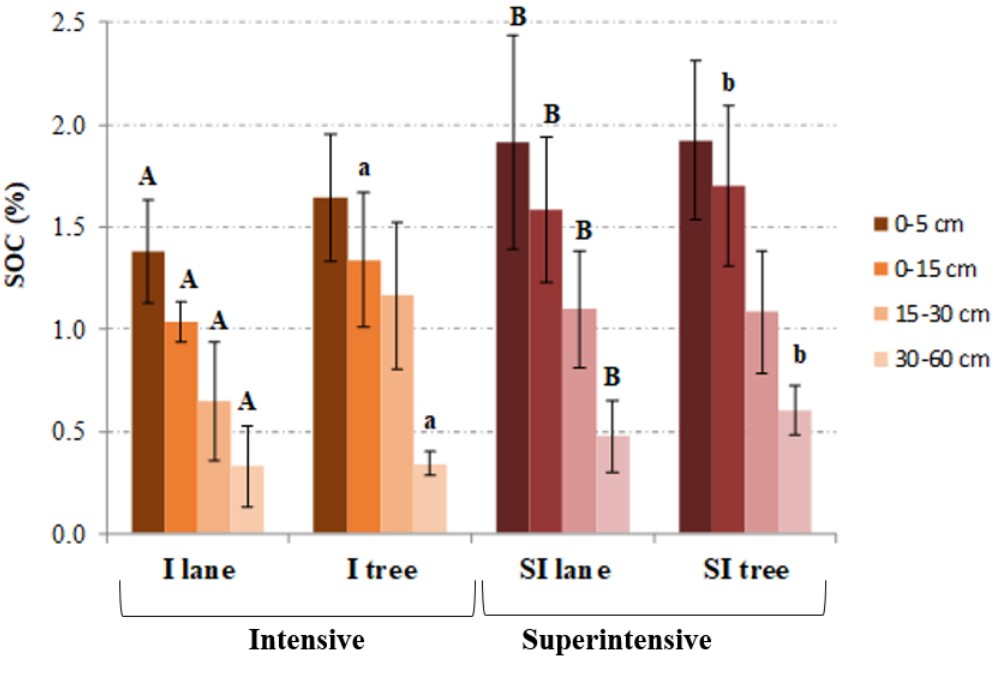

**(A)**

**Figure 4.** *Cont.*

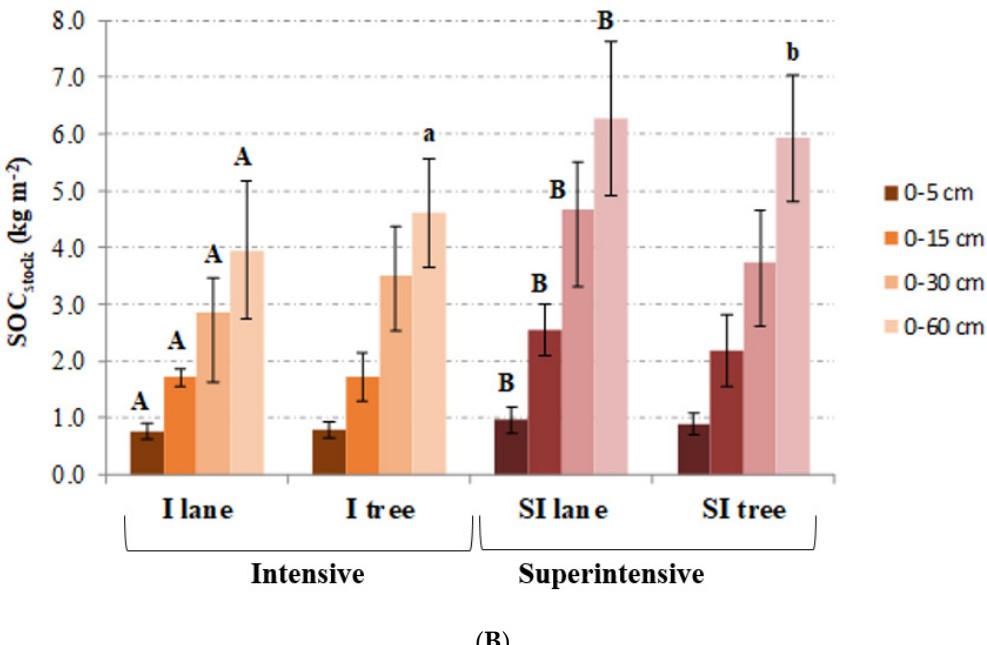

**(B)**

**Figure 4.** (**A**) Soil organic carbon concentration (SOC; %) and (**B**) cumulative soil organic carbon stock (SOC$_{stock}$; kg m$^{-2}$) by soil depth and location (lane or tree) within the two olive orchards (I: intensive; SI: superintensive) for the soil sampling carried out in 2018. Different letters indicate statistically significant differences ($p < 0.05$, using a Kruskal–Wallis test) between the two orchards for the same depth and location (lane, capital letters, or tree, lowercase letters). Error bars are standard deviation, n = 8.

*3.3. Comparison of SOC and SOC$_{stock}$ Evolution between 2013 and 2018 in Both Planting Systems*

Figure 5 compares the SOC measured in both orchards in 2013 and 2018 by location within the orchard and soil depth, thus allowing a more detailed discussion. In the intensive orchard (Figure 5A), there was no statistically significant increase in SOC in the top soil layer (0–5 cm). However, there was an overall increase in SOC in the top layer from 0–15 cm depth in both the lane and the tree areas. This increase was more intense in the tree area, which raised from 0.93 to 1.34% SOC, while in the lane area it increased from 0.79 to 1.04%. This higher rate of SOC increase in the tree area can be attributed to a higher biomass input into the soil, by tree roots and other plant material falling from the tree canopy, compared to the lane area. It is also interesting to note that in the intensive orchard, SOC in the soil surface (0–5 cm depth) seemed to be stabilized at around 1.4 and 1.6% in the lane and tree area, respectively. SOC evolution between 2013 and 2018 in the superintensive orchard (Figure 5B) presented a similar trend. On one side, as in the case of the intensive orchard, it did not show a significant increase in SOC in the top 0–5 cm soil layer, with values around 1.95% in the lane and tree areas. The increase in SOC was also concentrated in the top 0–15 cm depth, albeit it happened at a higher rate than that observed for the intensive orchard. SOC in the top 0–15 cm soil layer in the superintensive orchard increased, approximately, from 1.1 to 1.6% in the lane area and from 1.2 to 1.7% in the tree area.

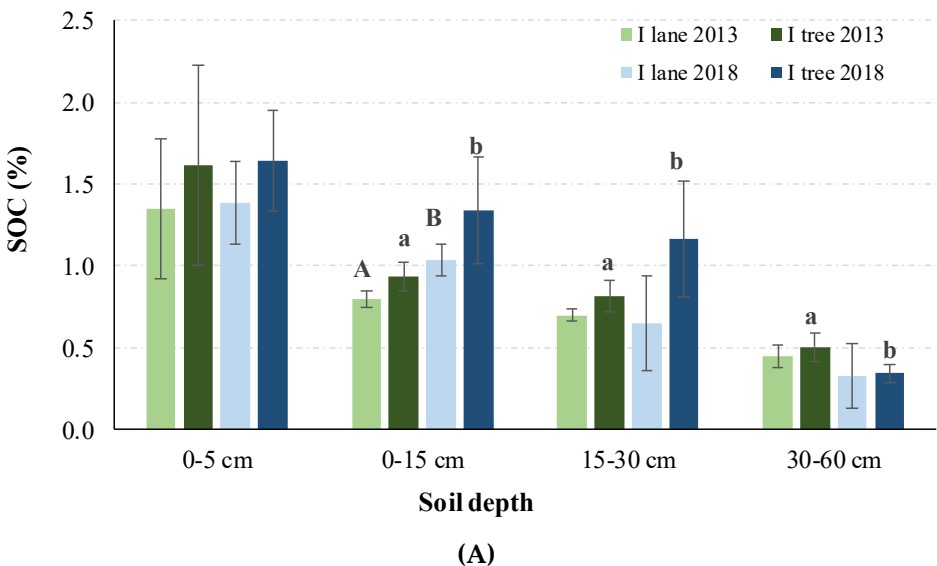

**(A)**

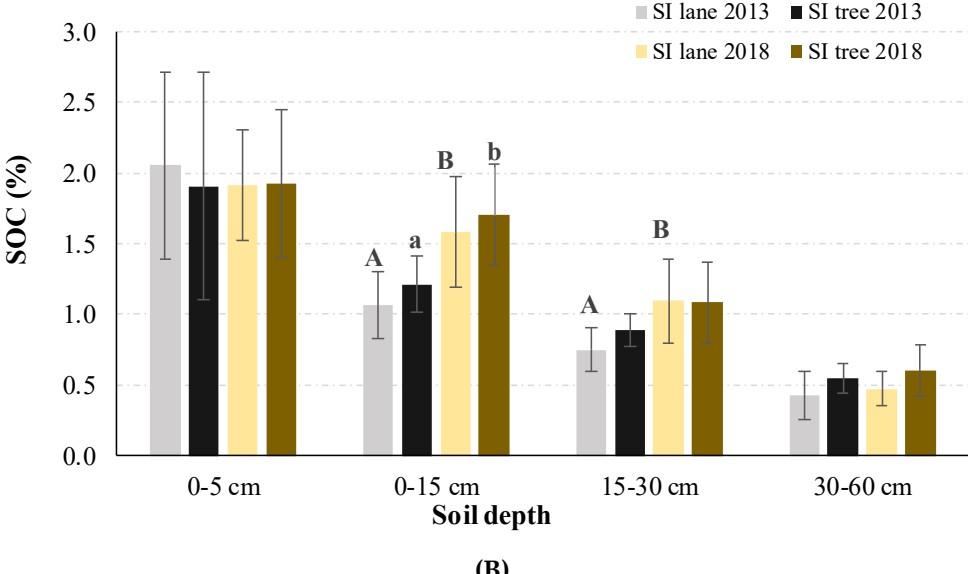

**(B)**

**Figure 5.** Comparison of soil organic carbon concentration (SOC; %) evolution between 2013 and 2018 in the intensive (**A**) and superintensive (**B**) olive orchard. Different letters indicate statistically significant differences ($p < 0.05$, using a Kruskal–Wallis test) between the two soil sampling dates for the same soil depth (0–5, 0–15, 15–30 and 30–60 cm) and location (lane, capital letters, or tree, lowercase letters) within each orchard. Error bars are standard deviation, n = 8.

Figure 6 allows comparison of the calculated $SOC_{stock}$ values measured in 2013 and 2018. In the case of the intensive orchard (Figure 6A), our measurements detected a statistically significant increase in the top 0–15 soil layer during this period. $SOC_{stock}$ up to a 15 cm depth increased from 1.3 to 1.7 kg m$^{-2}$, on average for the lane and tree areas. However, this increase could not prove to be statistically significant for the whole rooting depth (0–60 cm). Our interpretation is that the high spatial variability in our SOC measurements masks detection of differences between years, as the absolute differences in $SOC_{stock}$ decreases as we computed $SOC_{stock}$ by aggregating all the soil depths sampled. This was compounded by the lack of increase in SOC in the 30–60 cm depth layer, half of the soil rooting depth, which diluted the differences and further complicated statistical detection of differences. It is apparent that our results show that SOC accumulation has

kept increasing in the intensive orchard, at least in the 0–15 cm soil depth, but also that a larger number of replications are necessary to detect statistically significant variations in $SOC_{stock}$ under the conditions of many olive orchards in Mediterranean conditions. The superintensive orchard (Figure 6B) presented a clearer trend, detecting a statistically significant increase in the $SOC_{stock}$ for the 0–15, 0–30 and 0–60 soil depths in the lane and tree areas, which we believe it was possible for the higher increase in SOC between both dates, which allowed statistical detection even with a moderate number of samples. The change in $SOC_{stock}$ in 0–15 cm depth was, approximately, from 1.6 to 2.4 kg m$^{-2}$, on average for the lane and tree areas, and for the 0–60 cm depth from 4.7 to 6.1 kg m$^{-2}$, on average for the lane and tree areas.

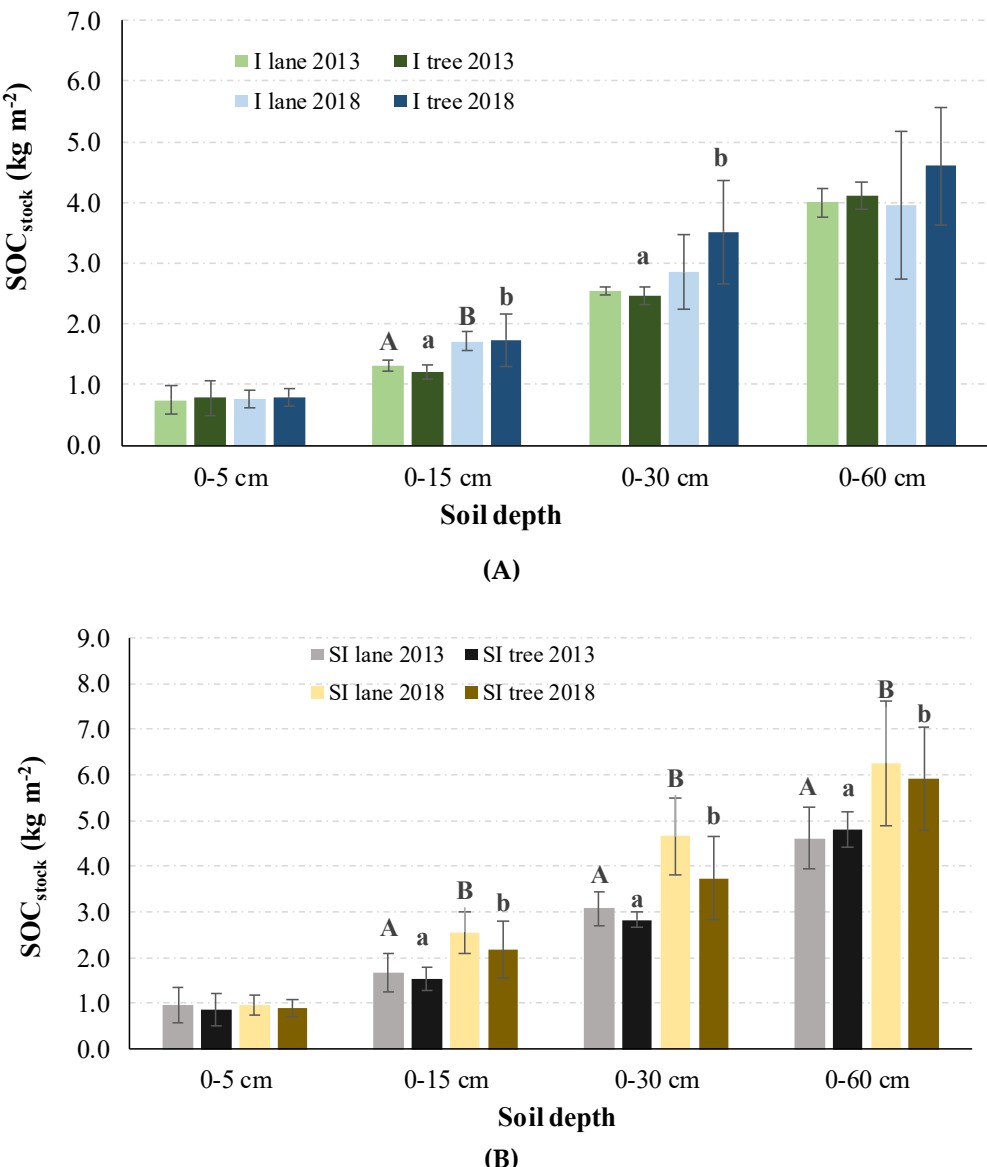

**Figure 6.** Comparison of cumulative soil organic carbon stock ($SOC_{stock}$; kg OC m$^{-2}$) evolution between 2013 and 2018 in the intensive (**A**) and superintensive (**B**) olive orchard. Different letters indicate statistically significant differences ($p < 0.05$, using a Kruskal–Wallis test) between the two soil sampling dates for the same soil depth (0–5, 0–15, 0–30 and 0–60 cm) and location (lane, capital letters, or tree, lowercase letters) within each orchard. Error bars are standard deviation, n = 8.

Between the two sampling dates, there was a large increase in SOC in the superintensive orchard, which, on average for all the orchard areas and soil depths, went from 0.84

to 1.09% between 2013 and 2018, and translated into an increase in SOC$_{stock}$ for the whole rooting depth 4.7 vs. 6.1 kg m$^{-2}$ in the superintensive orchard, around a 30% increase in a roughly 5-year period. This large increase, well above the "4 per 1000" [18], can be explained mainly by soil management and planting density. This has allowed a significant increase in the soil organic carbon content from a low level of soil organic carbon typical of many olive orchards in Mediterranean regions [3]. This was due to the high biomass return to the soil as a result of soil the management practices based on no till with a mulch of chopped pruning residues and the high tree density, which also results in high roots biomass return to the soil, facilitated by the deficit irrigation. In the case of the intensive orchard, our previous discussion suggests that SOC$_{stock}$ evolution merits analysis despite the inconclusive statistical results for the 0–60 cm soil depth. In the intensive orchard, the average SOC, averaged for all the areas and depths, went from 0.70 to 0.81% between 2013 and 2018, and translated into a change in the average SOC$_{stock}$ for the whole rooting depth 0–60 cm from 4.1 to 4.2 kg OC m$^{-2}$. This increase in approximately 2% is also close to the "4 per 1000" target [18]. However, until we developed better sampling and analytical strategies to enhance our capabilities to monitor these rates of moderate SOC increase in soils, we might be underestimating the potential for this kind of olive orchard to provide a net carbon sequestration in the soil, although monitoring soil organic carbon in the longer term (allowing to build higher absolute differences between dates) in reference orchards could be a useful strategy.

## 4. Conclusions

Duplicated soil sampling, spaced by, approximately, five years (2013 and 2018), in two mature, deficit-irrigated olive orchards in southern Spain, on the same soil type but with different planting systems, has allowed the quantification of its SOC and SOC$_{stock}$ for the full soil rooting depth, and to identify trends regarding its temporal evolution and spatial distribution. In both sampling dates, the intensive olive orchard presented a higher SOC in the tree area as compared to the lane area, but this did not result in a higher SOC$_{stock}$ in the tree area due to a higher bulk density in the lanes. The superintensive olive orchard did not present differences in the SOC between the two areas (lane vs. tree) for the two sampling dates.

The SOC in the soil increased significantly in the superintensive orchard during the 5-year period, from an average (for the whole soil profile) of 1.1 to 1.6% in the lane area, and from 1.2 to 1.7% in the tree area, resulting in a statistically significant increase in SOC$_{stock}$ in the orchard of approximately 30% for the 0–60 cm depth, from 4.7 to 6.1 kg m$^{-2}$. This increase in SOC and SOC$_{stock}$ was absent in the 0–5 cm topsoil, and it was most intense in the 0–15 cm soil depth, particularly in the tree area.

In the intensive orchard, there was a statistically significant increase in SOC$_{stock}$ in the top 0–15 cm of soil, but it did not detect a statistically significant increase for the top 0–60 cm, which presented an average of 4.1 and 4.2 kg OC m$^{-2}$ in 2013 and 2018, respectively. These results suggest there still has been an accumulation of SOC in the intensive orchard, but given its relatively moderate magnitude, it has still been difficult to detect without resorting to a large number of measurements to address the issue of spatial variability.

Overall, our results indicate that under typical Mediterranean conditions with a very limited, 0–60 cm, rooting depth, there is proven potential for a significant increase in organic carbon into the soil by olive orchards, in our case with soil management based in a mulch of chopped pruning residues, cover crop by natural vegetation present at the orchard and access to deficit irrigation, which allows higher vegetative growth by the trees. Under the conditions of our study, the superintensive olive orchard presented a higher rate of increase in SOC$_{stock}$ as compared to the intensive olive orchard. In both cases, the increase is associated with an increase in relatively deep soil layers, mainly 0–15 cm depth, while the SOC in the 0–5 cm topsoil layer did not show a significant increase. Our results also point to the need for improvement in our detection capabilities to efficiently monitor moderate increases in SOC, as in the case for intensive orchards. These results can provide insight

for current discussions on the actual potential of carbon sequestration by olive orchards, as well as on the provision of ecosystem services by different typologies of olive groves, in combination with other key factors related to a proper functioning soil (e.g., biological activity or physical structure).

**Author Contributions:** Conceptualization, M.-A.S. and J.A.G.; methodology, M.-A.S., J.A.G., P.F.R. and L.R.-B. soil sampling and analysis M.-A.S., J.A.G. and L.R.-B., formal analysis, M.-A.S. and J.A.G., writing original draft preparation, M.-A.S. and J.A.G., review and editing, M.-A.S., J.A.G., P.F.R. and L.R.-B.; funding acquisition, J.A.G., P.F.R. and L.R.-B. All authors have read and agreed to the published version of the manuscript.

**Funding:** This received funding by projects PID2019-105793RB-I00 and Severo Ochoa and María de Maeztu Program for Centers and Units of Excellence in R&D (Ref. CEX2019-000968-M) both Spanish Government, SHui (European Commission Grant Agreement number: 773903), TUdi (European Commission Grant Agreement number 101000224) and EU-FEDER funds.

**Data Availability Statement:** Ancillary data from the figures shown are available at DIGITAL CSIC https://digital.csic.es/handle/10261/263199 (accessed on 7 February 2022).

**Acknowledgments:** The authors would also like to thank to the owners of "El Alamillo" farm, D. Francisco Pérez and Daniel Pérez for his continuous support to our research in his farm.

**Conflicts of Interest:** The authors declare no conflict of interest.

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
