# Peer review of "Comparison of Soil Organic Carbon Stocks Evolution in Two Olive Orchards with Different Planting Systems in Southern Spain"

_agriculture, doi:10.3390/agriculture12030432_

Round 1

Reviewer 1 Report

The manuscript entitled “Comparison of soil organic carbon stocks evolution in two olive orchards with different planting system in Southern Spain" is based on original research experiment and the presented results therein broaden the knowledge in the field of applied plant science and horticultural science. Authors aimed to evaluate the differences in soil organic carbon (SOC) sequestration rate in the soil rooting depth between two olive orchards with different typologies, intensive and superintensive, in a period of five years. Moreover, they also aimed to appraise the effect of the differences in SOC concentration in the topsoil layer between the lane and the tree canopy area on the determination of SOCstock. The scope of work includes the performance of experiment in field conditions, during which broad soil analysis  was obtained.

There is no doubt that this work is in the scope of Agriculture journal. The publication presents generally interesting and important studies. The paper is well organized, presented in a logical sequence, and has adequate bibliographic review. The work delivers interesting results and can be the important source of valuable information.

The introduction is properly composed. The materials and methods section contains the basic requested elements and provide information about the experimental preparations, analyses and growth conditions. The data analysis is properly provided. The results show valuable information. The obtained data are discussed sufficiently.

However, authors made some shortcomings that must be corrected:

  • The abstract is very developed. It should be no longer than 200 words, and now it has more than 400. Moreover, at the beginning it does not outline the research problem.
  • Introduction: While I have no objections to the aims of the work, in the same paragraph I am missing a scientific hypothesis.
  • Table 1: It is not clear what the letters A, B and BC mean?
  • Figures 2, 4, 5, and 6: why were both small and capital letters used to denote homogeneous groups? If this is not a mistake, then you need to know the meaning of both in the description. The number of repetitions and S.E. or S.D. should also be given.

I would like to underline that my remarks are auxiliary and not undertake the quality and importance of the paper.

Author Response

Dear reviewer,

First we want to thank for your review and useful comments on the first version of our manuscript. The revised version, submitted with this letter, includes changes to address the modifications requested by the three reviewers. Below you can find our answer to each specific question raised by you, with your original remarks in Italics.

1) The abstract is very developed. It should be no longer than 200 words, and now it has more than 400. Moreover, at the beginning it does not outline the research problem.

We have revised the abstract shortening it to 210 words, and we have also outlined the research problem.

2) Introduction: While I have no objections to the aims of the work, in the same paragraph I am missing a scientific hypothesis.

We have edited the introduction to state clearly our scientific hypothesis, which was not well explained in the previous version of the manuscript.

3) Table 1: It is not clear what the letters A, B and BC mean?

Letters A, B and C refers to soil horizon from the soil profile description and this clarification has been added to a revised caption for Table 1.

4) Figures 2, 4, 5, and 6: why were both small and capital letters used to denote homogeneous groups? If this is not a mistake, then you need to know the meaning of both in the description. The number of repetitions and S.E. or S.D. should also be given.

We have revised Figures 2, 5, 5 and 6 to follow the reviewer comment. We have added the SD bars, eliminated the letters in groups with no differences and indicated the number of replications in the Figure caption.

Reviewer 2 Report

General comments:

The manuscript described the soil organic carbon stocks in two olive orchards with different planting system in southern Spain. It is interesting to explore  the effects of tree density and age on soil organic carbon. However, the result interpretation and discussion are limited. There is lack of mechanical explanation and evaluation of the results.

Specific comments:

in abstract, "in subsurface soil, mainly 0-15 cm depth" ??

Soil depths considered were 0-5, 0-15, 15-30 and 30-60 cm. Why is 0-15 cm not 10-15 cm?

Figure 3 is not clear.

Author Response

Dear reviewer,

First we would like to thank for your review and useful comments on the first version of our manuscript. This revised version includes the changes necessary to address the modifications requested by the three reviewers. Below you can find our answer to each specific question raised by you following your original remark in Italics.

1) in abstract, "in subsurface soil, mainly 0-15 cm depth" ??

We have edited the abstract to clarify this sentence.

2) Soil depths considered were 0-5, 0-15, 15-30 and 30-60 cm. Why is 0-15 cm not 10-15 cm?

It was made in this way because the coring system used (by hand in the top 0-5 cm) and hydraulic for the other depths in the whole soil profile (0 to 60 cm) did not allowed the sampling of cores (less than 10 cm) without increasing the risk of contamination by soil corresponding to different sampling depths The revised version of the manuscript has been edited to clarify this point.

3) Figure 3 is not clear.

Figure 3 has been revised to make it more clear, adding error bars with standard deviation, noting the number of replications (as suggested by reviewer 1), and changing the colors of the columns.

Reviewer 3 Report

The paper by Gomez et al. is well written. However, the underlying question and approach are so simple that it misses many answers.

The entire study comes down to two sets of soil cores collected 55 months apart. The cores were collected under two olive plantation management plans. They look at soil organic carbon as the significant variable. They collect samples under the trees and, for some reason, in the lanes.

As a soil scientist, I am unclear on the rationale for recovering the cores in the lane. (We avoid trafficked areas at all costs.) They are trafficked as the bulk density is up. I suggest their transects went in the wrong direction for any comparable information to be collected. We have no idea what the site variability looks like. Vehicle traffic (tree shakers) is not controlled. Gravel is sometimes added. Applications of chemicals and water (irrigation) would be different. Were the lanes only made in place or was soil added or mounded up? Where in the lane is the sample collected?

Overall the study shows little new information. First, 55 months is not long enough to show changes in the soil carbon at the level they indicate. Second, why are we concerned about the lanes as no clear rationale is provided. Their graphics are very confusing; why have capital and lower case letters? I know why but this needs to be explained better? Why do they cut the profiles up into four segments?

Figures 2 and 4 are just repeats of data over time. They do not tell us what goes into these sites. The entire project comes down to figure 6, a combination of 2, 4, and 5. Super-intensive management will increase soil carbon stocks (carbon x bulk density). The project hangs on the 0 to 60 cm samples (figure 6b.) Does this project suggest that we should increase the number of intensive lanes to increase soil carbon as the overall soil carbon levels are really not changing that much?

What about soil biology and other indicators? What about the infiltration of water? What about the other 15 things that go into a well-functioning soil?

Author Response

Dear reviewer,

First we would like to express our gratitude for your review and the useful comments on the first version of our manuscript. This revised version, accompanying this letter, includes the changes made in order to the modifications and clarifications requested by the three reviewers. Below you can find our answer to each specific question raised by you, following each of your remarks in Italics.

1) The entire study comes down to two sets of soil cores collected 55 months apart. The cores were collected under two olive plantation management plans. They look at soil organic carbon as the significant variable. They collect samples under the trees and, for some reason, in the lanes.

We collected samples in both, lane and tree, areas because prior studies noted that bulk density and soil organic carbon concentration might differ between these two areas, as noted in the introduction. We have edited the manuscript to express this reason in a clearer way.

2) As a soil scientist, I am unclear on the rationale for recovering the cores in the lane. (We avoid trafficked areas at all costs.) They are trafficked as the bulk density is up. I suggest their transects went in the wrong direction for any comparable information to be collected. We have no idea what the site variability looks like. Vehicle traffic (tree shakers) is not controlled. Gravel is sometimes added. Applications of chemicals and water (irrigation) would be different. Were the lanes only made in place or was soil added or mounded up? Where in the lane is the sample collected?

We collected soil samples in the lane area because SOC and bulk density might differ from those of the tree area (for some of the reasons indicated in your comments) based on previous studies, see answer to your comment above, and so, in our understanding, the only way to provide a comprehensive determination of the SOC stock is sampling both area. This is the reason also why the transects were made in the direction used. We have also revised the text to clarify the three questions noted in your comment:

1- Gravel was not added and that the lanes were made in place without making mounds where to plant the trees.

2- The samples in the lane were spread along the lane.

3) Overall the study shows little new information. First, 55 months is not long enough to show changes in the soil carbon at the level they indicate. Second, why are we concerned about the lanes as no clear rationale is provided. Their graphics are very confusing; why have capital and lower case letters? I know why but this needs to be explained better? Why do they cut the profiles up into four segments?

We clearly recognize that this is a modest study but, at the same time, it contributed to fill a gap indicated in the introduction, which is to enlarge the number of studies aimed to provide quantitative information on the effect of plant density in olive orchards (albeit in an specific conditions) on SOC concentration and SOC stock. As noted in the introduction, to our knowledge only one study (Lopez-Bellido et al., 2016) has attempted that but not for the whole soil rooting depth (A and B horizon).

Although high, and to us a surprise in its magnitude, the increase in SOCstock observed in the superintensive orchards is feasible according to previous studies. Our annual increase, 2.4 t ha-1 year-1 is in the range reported by Bellido et al. (2016) and also similar in the increase noted by Xiloyannis et al. (2018) in a traditional Italian olive orchard. It is less than 10% of the annual biomass return in an olive orchard with cover crops and use of mulch according to the estimation of carbon produced by the different elements of the olive orchard made by Xiloyannis (2015). We have edited the revised version of the manuscript to further clarify this point.

We acknowledge that the figures 2 to 6 might present some overlap among them, but we decided to use this format to facilitate the presentation and discussion of the results. For this reason, and also because the other two reviewers have not raised objections to their use we have decided to maintain them in the revised version of the manuscript. We also acknowledge that in the original version of the manuscript these Figures are a bit unclear. We have revised them following the indications of reviewer 1. The reason why the soil profile sampling was made at four depths (0-5, 0-15, 15-30 and 30-60) has been explained in our answer to reviewer 2 (below in quotation marks) and clarified in the revised version of the manuscript. “It was made this way because the coring system used (by hand in the top 0-5 cm) and hydraulic for the other depths in the whole soil profile (0 to 60 cm) did not allowed the sampling of small cores (less than 10 cm deep) without increasing the risk for contamination by soil corresponding to different sampling depths. The revised version of the manuscript has been edited to clarify this point.”

Lopez-Bellido, P.J.; Lopez-Bellido, L.; Fernandez-Garcia, P.; Muñoz-Romero, V.; Lopez-Bellido, F.J. Assessment of carbon se-questration and the carbon footprint in olive groves in Southern Spain, Carbon Manag. 2016, 7, 161-170, DOI: 10.1080/17583004.2016.1213126

Xiloyannis, C. CO2 storage in the soil. International course on the carbon footprint of olive growing. Madrid. 2015. Available at: https://oliveclima.eu/wp-content/uploads/2013/04/2015-Xiloyannis-COI-Madrid_p.pdf

Xiloyannis, C., Palese, A.M. Sofo, A., Minini, A.N. Lardo, E. Acta Hortic. 1199. The agro-ecosystemic benefits of sustainable management in an Italian olive grove. Acta Horticulturae ISHS 2018. Proceedings of the VIII International Olive Symposium. DOI 10.17660/ActaHortic.2018.1199.47

4) Figures 2 and 4 are just repeats of data over time. They do not tell us what goes into these sites. The entire project comes down to figure 6, a combination of 2, 4, and 5. Super-intensive management will increase soil carbon stocks (carbon x bulk density). The project hangs on the 0 to 60 cm samples (figure 6b.) Does this project suggest that we should increase the number of intensive lanes to increase soil carbon as the overall soil carbon levels are really not changing that much?

We acknowledge that the figures 2 to 6 might present some overlap among them, but we decided to use this format to facilitate the presentation and discussion of the results. For this reason, and also because the other two reviewers have not raised objections to their use we have decided to maintain them in the revised version of the manuscript.

The results presented in the manuscript notes that olive orchards with a higher number of trees, providing that they also use a cover crops and mulch of pruning residues, results in a higher amount of organic carbon stored in the soil as compared to a lower plant density. The manuscript notes that this should also considered when discussing provision of ecosystem services by different types of olive orchards. We do not present in the manuscript a simple claim, since the plant density depends on several factors such as soil properties, water availability, topography, … We understand that the manuscript conclusion in its revised version presents this idea clearly.

5) What about soil biology and other indicators? What about the infiltration of water? What about the other 15 things that go into a well-functioning soil?

Those are very interesting questions and have been studied by our team for a long time as it can be seen in the revised article. However, the scope of our study was not to study all the elements related to soil functions, it was something more modest. Our objective was to provide some quantitative insight into one of the elements of interest in a functional soil, evolution of soil organic carbon according to planting density. We have edited the manuscript to make this point more clear, in the introduction. Also to avoid any possible confusion to mix our conclusion with a general claim on soil functions.  

Round 2

Reviewer 2 Report

The revised manuscript was improved sufficiently. 

Reviewer 3 Report

The design of this study is extremely weak. Nothing can be done about it now but think about it in the future,